# Emergence of $\mathcal{PT}$-symmetry breaking in open quantum systems

J. Huber[1*], P. Kirton[1,2], S. Rotter[3], P. Rabl[1],

**1** Vienna Center for Quantum Science and Technology, Atominstitut, TU Wien, 1040 Vienna, Austria
**2** Department of Physics and SUPA, University of Strathclyde, Glasgow G4 0NG, UK
**3** Institute for Theoretical Physics, TU Wien, 1040 Vienna, Austria
* julian.huber@tuwien.ac.at

May 29, 2020

## Abstract

The effect of $\mathcal{PT}$-symmetry breaking in coupled systems with balanced gain and loss has recently attracted considerable attention and has been demonstrated in various photonic, electrical and mechanical systems in the classical regime. However, it is still an unsolved problem how to generalize the concept of $\mathcal{PT}$ symmetry to the quantum domain, where the conventional definition in terms of non-Hermitian Hamiltonians is not applicable. Here we introduce a symmetry relation for Liouville operators that describe the dissipative evolution of arbitrary open quantum systems. Specifically, we show that the invariance of the Liouvillian under this symmetry transformation implies the existence of stationary states with preserved and broken parity symmetry. As the dimension of the Hilbert space grows, the transition between these two limiting phases becomes increasingly sharp and the classically expected $\mathcal{PT}$-symmetry breaking transition is recovered. This quantum-to-classical correspondence allows us to establish a common theoretical framework to identify and accurately describe $\mathcal{PT}$-symmetry breaking effects in a large variety of physical systems, operated both in the classical and quantum regimes.

# 1 Introduction

The breaking of parity and time-reversal ($\mathcal{PT}$) symmetry has been widely studied in dissipative systems with an exact balance between gain and loss [1–5]. Owing to this symmetry, the dynamical matrix describing such systems may exhibit a purely real eigenvalue spectrum, despite a constant exchange of energy with the environment. As the dissipation rates are increased above a critical value, at least one pair of eigenvalues becomes purely imaginary and the corresponding eigenmodes no longer exhibit the symmetry of the underlying equations of motion. Over the past years, this effect has attracted considerable attention and has been demonstrated in various optical [6–8], electrical [9] and mechanical [10] settings. Apart from purely fundamental interest, this mechanism also has many important practical consequences, for example, for the operation of multi-mode lasers [11, 12], enhanced measurements [13–18], the bandstructure of dissipative lattice systems [19] or energy transport at macroscopic [20] and microscopic [21, 22] scales.

In connection with $\mathcal{PT}$-symmetric systems it is common to use the terminology of non-Hermitian 'Hamilton operators'. However, the effect described above is *a priori* only defined for classical systems that can be modeled in terms of a complex-valued dynamical matrix [23]. Indeed, in a full master equation (ME) formulation of open quantum system [25], there is no such transition between purely real and purely imaginary eigenvalues of the corresponding Liouville operator. Also, at a microscopic level, the time-reversal equivalence between loss and gain is broken by quantum fluctuations [26–29]. Therefore, it is still an unresolved question how to formally define $\mathcal{PT}$-symmetry for dissipative quantum systems [30] and if the breaking of this symmetry can exist at all at a microscopic level [29]. In several previous studies this question has been addressed by looking at coupled quantum oscillators [17, 26–29, 31–36] or bosonic atoms [37] with gain and loss, or at equivalent coherent, but unstable systems [38]. In such settings, the symmetry-breaking effect can still be observed in the dynamics of the mean amplitudes, which simply reproduce the classical equations of motion, while quantum effects lead to increased fluctuations. However, these findings cannot be generalized to systems with a finite dimensional Hilbert space and they also provide no insigths into the stationary states of $\mathcal{PT}$-symmetric quantum systems [21, 28], which do not exist in purely linear models.

In this work we introduce a symmetry transformation for Liouville operators, which extends the conventional definition of $\mathcal{PT}$ symmetry to arbitrary open quantum systems. We show that under very generic conditions, the existence of this symmetry implies that the steady state of the system can be tuned between a fully symmetric and a symmetry-broken phase. While the change from one to the other limiting state is always continuous, it be-

comes more and more pronounced as the dimension of the Hilbert space is increased, and a sharp $\mathcal{PT}$-symmetry breaking transition emerges in the semiclassical limit. This quantum-to-classical correspondence allows us to establish a unified theoretical framework for analyzing $\mathcal{PT}$-symmetry breaking effects in a wide range of physical systems and to identify characteristic properties and experimentally observable features that are common to all of them.

## 2 $\mathcal{PT}$-symmetric quantum systems

We consider a generic bipartite quantum system with total Hamiltonian $H$. The two subsystems, A and B, have the same Hilbert space dimension, $d$, and are subject to dissipation described by the local jump operators $c_A$ and $c_B$, respectively. The ME for the system density operator $\rho$ can then be written as ($\hbar = 1$) [25]

$$\dot{\rho} = -i\left(H_{\text{eff}}\rho - \rho H_{\text{eff}}^{\dagger}\right) + c_A\rho c_A^{\dagger} + c_B\rho c_B^{\dagger} \equiv \mathcal{L}\rho, \tag{1}$$

where $\mathcal{L} \equiv \mathcal{L}[H; c_A, c_B]$ is the Liouvillian superoperator. The first term in Eq. (1) describes the evolution of a quantum state under the action of the non-Hermitian Hamiltonian

$$H_{\text{eff}} = H - \frac{i}{2}c_A^{\dagger}c_A - \frac{i}{2}c_B^{\dagger}c_B. \tag{2}$$

This part does not conserve the norm of the state and thus the recycling terms $\sim c\rho c^{\dagger}$ must be added to obtain a trace-preserving dynamics.

Given the decomposition of a ME in Eq. (1), it is tempting to define $\mathcal{PT}$-symmetric quantum systems in analogy to the classical case [4, 5], namely as open quantum systems where $(\mathcal{PT})H_{\text{eff}}(\mathcal{PT})^{-1} = H_{\text{eff}}$. Here $\mathcal{P}$ is the parity operator with $\mathcal{P}(A \otimes B)\mathcal{P}^{-1} = B \otimes A$ and $\mathcal{T}i\mathcal{T}^{-1} = -i$. However, $H_{\text{eff}}$ has only negative imaginary parts because the norm of a state evolving under $H_{\text{eff}}$ always decreases. Thus, this symmetry relation can only be satisfied in closed systems. The same is then also true for the eigenvalues of the full Liouville operator $\mathcal{L}$ whose real part must always be negative or zero. In conclusion, while there is a natural way to introduce non-Hermitian Hamiltonians in open quantum systems and even probe them via conditional measurements [39–43], there are no $\mathcal{PT}$-symmetric (super-)operators in the conventional sense.

To extend the concept of $\mathcal{PT}$ symmetry into the quantum regime, it is important to keep in mind that the relevant physical effect of the $\mathcal{T}$-operator is to exchange loss and gain and not to implement a time-reversal transformation. While in the classical case both operations are equivalent and usually no distinction is made, this is no longer true for quantum systems. In the simplest example of a quantum harmonic oscillator the effect of loss with rate $\Gamma$ is modeled by a jump operator $c = \sqrt{\Gamma}a$, where $a$ is the annihilation operator. In turn, the effect of gain with the same rate can be described by modifying the jump operator to be $c = \sqrt{\Gamma}a^{\dagger}$. Therefore, in this case we find that the transformation between loss and gain is implemented in the ME formalism by replacing the jump operator by its adjoint, $c \rightarrow c^{\dagger}$.

Guided by this explicit example, we introduce the following anti-unitary transformation for operators $O$,

$$\mathbb{PT}(O) = \mathcal{P}O^{\dagger}\mathcal{P}^{-1}, \tag{3}$$

and define an open quantum system to be $\mathcal{PT}$-*symmetric*, if the corresponding Liouvillian satisfies

$$\mathcal{L}[\mathbb{PT}(H); \mathbb{PT}(c_A), \mathbb{PT}(c_B)] = \mathcal{L}[H; c_A, c_B]. \tag{4}$$

This condition implies that the Hamiltonian $H$ is parity-symmetric and that the local jump operators are of the form

$$c_A = \sqrt{\Gamma} O \otimes \mathbb{1}, \qquad c_B = \sqrt{\Gamma} \mathbb{1} \otimes O^\dagger, \tag{5}$$

where $O$ can be an arbitrary dimensionless operator. We remark that this definition differs from the $\mathcal{PT}$-symmetric Liouville operators introduced in Ref. [30], where, to our knowledge, the considered transformations have no immediate physical interpretation or classical correspondence. While the systems studied in Refs. [30, 44, 45] satisfy Eq. (4) with a redefinition of $\mathcal{P}$, none of the examples discussed below exhibits the symmetry considered in these references when $d > 2$.

## 3 Phenomenology

Before we return to a more general discussion of Eq. (4), let us illustrate its physical implications in terms of two simple examples: (i) Two coupled spin $S = (d-1)/2$ systems with $O = S^+$, where $S^+ = S^x + iS^y$ is the spin raising operator, and (ii) two coupled harmonic oscillators with $O = a^\dagger$. In the second example we introduce a finite cutoff occupation number, i.e., $a^\dagger |n = d - 1\rangle = 0$. This cutoff mimics the effect of saturation in realistic systems [21] and allows us to vary the Hilbert space dimension. In both examples we consider a Hamiltonian of the form

$$H = g(O_A O_B^\dagger + O_A^\dagger O_B), \tag{6}$$

where $O_A = O \otimes \mathbb{1}$ and $O_B = \mathbb{1} \otimes O$. This Hamiltonian describes the coherent exchange of energy between the two subsystems with a strength $g$. The resulting Liouvillian, $\mathcal{L}[H; \sqrt{\Gamma} O_A, \sqrt{\Gamma} O_B^\dagger]$, then satisfies Eq. (4).

We calculate the steady state, $\rho_0$, satisfying $\mathcal{L}\rho_0 = 0$, for different ratios $\Gamma/g$ and show in Fig. 1 the symmetry parameter [28]

$$\Delta = \frac{|\langle O_A^\dagger O_A - O_B^\dagger O_B\rangle|}{\langle O_A^\dagger O_A + O_B^\dagger O_B\rangle} \leq 1. \tag{7}$$

This is an experimentally observable quantity, only requiring measurements of local operators, which provides a measure for the symmetry of the system, i.e., $\Delta = 0$ for a parity-symmetric density operator, $\mathcal{P}\rho\mathcal{P}^{-1} = \rho$. For the current examples, $\Delta$ represents the normalized population imbalance between the two subsystems. For small dimensions $d$, this parameter changes gradually from 0 to 1 with increasing $\Gamma$. This smooth variation is expected since observables of finite dimensional quantum systems cannot exhibit any non-analytic behavior. However, as the system size increases, $\Delta$ vanishes for $\Gamma/g < 1$ in the limit $d \to \infty$, while it retains a finite value for $\Gamma/g > 1$. In both examples, the critical ratio is $\Gamma/g = 1$, which corresponds to the dynamical $\mathcal{PT}$-symmetry breaking point of an equivalent linear oscillator system with gain and loss [4, 5]. We thus conclude that $\mathcal{PT}$-symmetry breaking, i.e., a non-analytic transition between two steady states with different symmetries, exists even for non-harmonic and finite dimensional quantum systems, but only as an emergent phenomenon in the semiclassical limit.

To obtain better insights into the nature of the two phases, we plot in Fig. 2(a) the purity, $P = \text{Tr}\{\rho_0^2\}$, for the steady state of the spin system. This quantity again exhibits a sharp transition around $\Gamma = g$ and shows that the symmetric and symmetry-broken phases

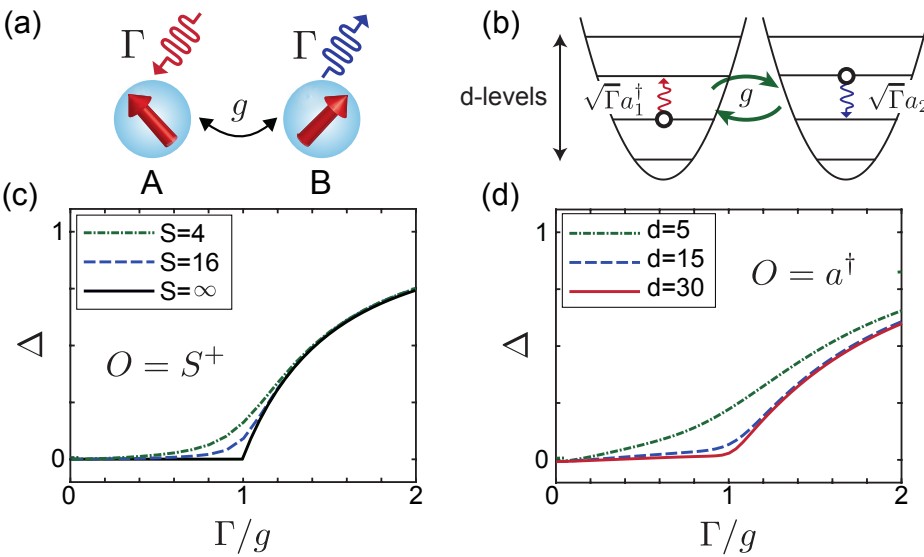

Figure 1: Two basic examples of $\mathcal{PT}$-symmetric quantum systems with a finite Hilbert space dimension $d$: (a) two coupled spin $S = (d-1)/2$ systems and (b) two coupled harmonic oscillators with a finite number of energy levels. In (c) and (d) we plot the corresponding dependence of the symmetry parameter $\Delta$ defined in Eq. (7) on the ratio $\Gamma/g$. In (c) the line for $S = \infty$ is obtained from a Holstein-Primakoff approximation (see discussion of Eq. (13)).

are characterized by a highly mixed and an almost pure steady state, respectively. More precisely, the scaling $P(\Gamma \to 0) \simeq d^{-2}$ implies that in the symmetric phase the steady state is close to the maximally mixed state, $\rho_0(\Gamma \ll g) \simeq \mathbb{1}/d^2$. This indicates that for $\Gamma < g$ the gain and loss processes cancel out on average while quantum fluctuations still occur with rate $\Gamma$ and completely randomize the system's long-time dynamics [21, 28]. In contrast, for $\Gamma > g$, the incoherent processes dominate and pump the spins into the polarised pure state, $\rho_0(\Gamma \gg g) \simeq |\psi_0\rangle\langle\psi_0|$, which satisfies $O_A|\psi_0\rangle = O_B^\dagger|\psi_0\rangle = 0$. Closer to the transition point, the coherent coupling creates excitations $\sim O_A^\dagger O_B|\psi_0\rangle$ on top of this state, which are strongly correlated. As shown in Fig. 2(b), this results in a characteristic peak in the entanglement negativity $\mathcal{N}$ around the transition point, which is a measure of non-classical correlations between the two subsystems [46, 47]. These correlations vanish again in the symmetric phase due to fluctuations. Consistent with similar features observed in saturable oscillator systems [21], this peak in the entanglement shows that even for $d \gg 1$ the $\mathcal{PT}$-symmetry breaking transition retains genuine quantum mechanical properties.

## 4 Existence of a fully symmetric steady state

We will now show that the properties discussed above for specific examples are indeed a general consequence of the symmetry relation in Eq. (4). Firstly, we demonstrate that, for any Liouvillian that satisfies this condition and where the spectrum of $H$ is non-degenerate, the fully mixed state,

$$\rho_0(\Gamma \to 0^+) = \frac{\mathbb{1}}{d^2}, \tag{8}$$

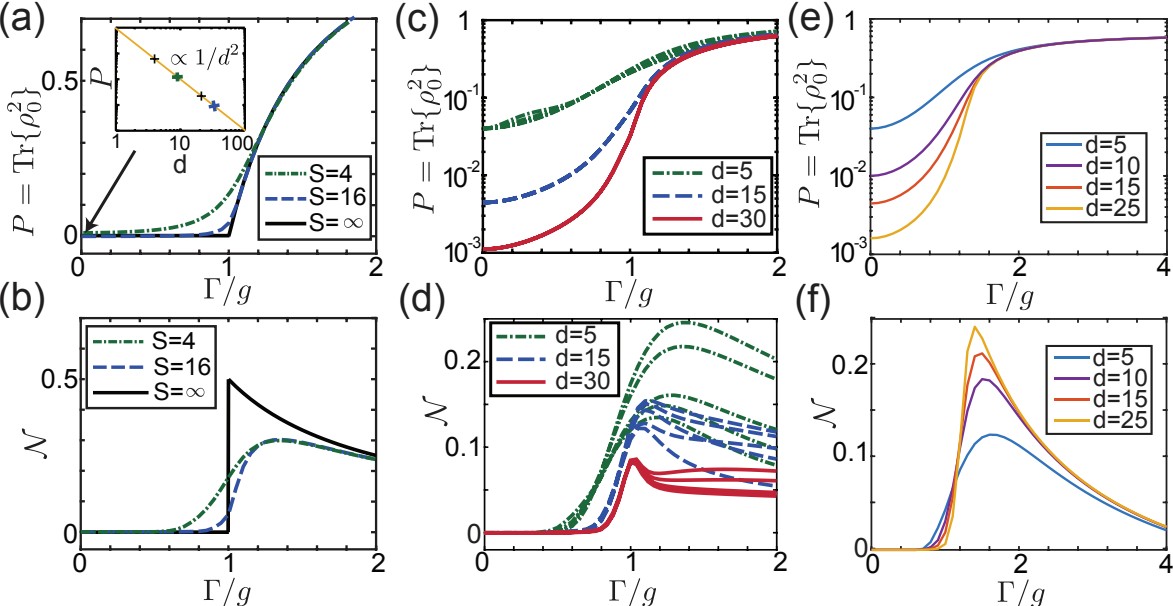

Figure 2: (a) Plot of the purity $P$ of the steady state of a $\mathcal{PT}$-symmetric spin dimer [see Fig. 1(a)] as a function of the dissipation rate and for different values of $S$. The inset shows that the purity satisfies $P \simeq 1/d^2$ for $\Gamma \ll g$. (b) Plot of the entanglement negativity $\mathcal{N}$ [46,47] for the same model. In (c) and (d) the same quantities are plotted for $\mathcal{PT}$-symmetric systems with random jump operators, as described in Appendix B, and in (e) and (f) for the generalized spin model defined in Sec. 6.

is a stationary state of $\mathcal{L}$ in the limit of a vanishingly small, but finite $\Gamma$. To do so we decompose $\mathcal{L} = \mathcal{L}_H + \mathcal{L}_\Gamma$, where $\mathcal{L}_H \rho = -i[H, \rho]$ describes the coherent evolution and $\mathcal{L}_\Gamma \rho = \sum_{\eta=A,B}(2c_\eta \rho c_\eta^\dagger - c_\eta^\dagger c_\eta \rho - \rho c_\eta^\dagger c_\eta)/2$. Further, we write the density operator in the eigenbasis of $H$ as

$$\rho = \sum_{n,m} \rho_{n,m}|E_n\rangle\langle E_m|, \tag{9}$$

where $H|E_n\rangle = E_n|E_n\rangle$. For $\Gamma = 0$ any diagonal state with $\rho_{n,m} = 0$ for $n \neq m$ is a stationary solution of the ME, but the populations $p_n = \rho_{n,n}$ are not uniquely determined. To show that for small but finite $\Gamma$ only the fully mixed state is dynamically stable, we write $p_n = 1/d^2 + \delta p_n$. Up to first order in $\Gamma$ we then obtain

$$\delta \dot{p}_n = \frac{1}{d^2}\langle E_n|\left([c_A, c_A^\dagger] + [c_B, c_B^\dagger]\right)|E_n\rangle. \tag{10}$$

We now make use of the relation $\mathcal{P}c_B\mathcal{P}^{-1} = c_A^\dagger$, which follows from Eq. (5), and the fact that the eigenstates of $H$ are also eigenstates of the parity operator, i.e., $\mathcal{P}|E_n\rangle = \pm|E_n\rangle$. This allows us to rewrite

$$\begin{aligned}\langle E_n|[c_B, c_B^\dagger]|E_n\rangle &= \langle E_n|\mathcal{P}[c_B, c_B^\dagger]\mathcal{P}^{-1}|E_n\rangle \\ &= -\langle E_n|[c_A, c_A^\dagger]|E_n\rangle,\end{aligned} \tag{11}$$

and $\delta \dot{p}_n = 0$. This result shows that for $\mathcal{PT}$-symmetric quantum systems the fully mixed state is stationary in the presence of a small amount of dissipation, even when each individual jump operator $c_{A,B}$ would drive the system into a polarized state. Note that the analysis presented here assumed a non-degenerate spectrum, i.e., the absence of any additional symmetries, $\mathcal{S}$, other than parity. In the general case the same arguments still hold as long as all eigenstates with different parity are separated by a finite energy gap, or, more formally, as long as $[\mathcal{S}, \mathcal{P}] = 0$. For a detailed proof and explanation see Appendix A.

## 5 Symmetry-breaking transition

While the existence of a fully symmetric steady state follows directly from Eq. (4), there are many trivial cases where this is also the only stationary state, for example, when $O$ is Hermitian. Therefore, we are interested in systems where there is a competing asymmetric phase in the limit $\Gamma \to \infty$. To ensure that such a phase exists we now restrict ourselves to a Hamiltonian as given in Eq. (6) and a non-Hermitian jump operator of rank $d - 1$ with $\text{Tr}\{O\} = 0$. This implies that there are dark states $|D\rangle$ and $|D^*\rangle$, which satisfy $O|D\rangle = 0$ and $O^\dagger|D^*\rangle = 0$. Under these assumptions we obtain the symmetry-broken phase

$$\rho_0(\Gamma \to \infty) = |D\rangle\langle D| \otimes |D^*\rangle\langle D^*|, \tag{12}$$

which is fully asymmetric, $\Delta = 1$, and has maximal purity, $P = 1$. Note, however, that for observing symmetry-breaking effects it is not essential that $\rho_0(\Gamma \to \infty)$ is a pure state and, later in this manuscript, we discuss examples where the symmetry-broken state is mixed.

Given the two distinct limiting phases, the remaining question is, if there is a sharp phase transition between them at a critical intermediate value $\Gamma_c$. For the spin system discussed above this question can be rigorously answered in the limit $S \gg 1$ by examining the stability

of linear fluctuations on top of the fully polarized state. This can be done using a Holstein-Primakoff approximation [48], where the spin operators are replaced by a pair of bosonic operators, $S_A^- \simeq \sqrt{2S}\, a^\dagger$, $S_A^+ \simeq \sqrt{2S}\, a$, $S_B^- \simeq \sqrt{2S}\, b$ and $S_B^+ \simeq \sqrt{2S}\, b^\dagger$, where $[a, a^\dagger] = [b, b^\dagger] = 1$. This approximate transformation brings the ME into a quadratic form,

$$\dot{\rho} = -i[H_{\text{lin}}, \rho] + \Gamma \mathcal{D}[a]\rho + \Gamma \mathcal{D}[b]\rho, \tag{13}$$

with Hamiltonian $H_{\text{lin}} = g(ab + a^\dagger b^\dagger)$. From the analytic solution of this linearized model we find that the fluctuations $\langle a^\dagger a \rangle$ and $\langle b^\dagger b \rangle$ diverge at the point $\Gamma_c = g$. Explicitly, in terms of the original spin expectation values we obtain

$$\langle S_{A/B}^z \rangle_0 = \pm S \mp \frac{g^2}{2(\Gamma^2 - g^2)}. \tag{14}$$

Similarly, we can use well-known results for Gaussian states [49] and derive analytic expressions for the purity and the entanglement negativity,

$$P = 1 - \frac{g^2}{\Gamma^2}, \qquad \mathcal{N} = \frac{g}{2\Gamma}. \tag{15}$$

These predictions are shown as the curves labeled by $S \to \infty$ in Fig. 2(a)–(b). Within this Holstein-Primakoff approximation, the substantial amount of entanglement with a maximum of $\mathcal{N}(\Gamma = \Gamma_c) = 1/2$ at the transition point can be directly understood from the form of $H_{\text{lin}}$, which represents a two-mode squeezing interaction.

In general, such an analytic treatment is not possible and, in many situations, $\mathcal{PT}$-symmetry breaking can occur as a smooth crossover, rather than a sharp phase transition. Nevertheless, it turns out that the appearance of a sharp transition in the limit of large $d$ does not require any specific fine tuning of the dissipation mechanism. This point is illustrated in Fig. 2(c)–(d), where we consider a set of $\mathcal{PT}$-symmetric quantum systems with randomly generated jump operators $O$. For each individual line in this plot a jump operator $O$ has been constructed by a Cholesky decomposition of an operator $R = OO^\dagger$, which is drawn randomly from the Gaussian orthogonal ensemble (GOE) (see Appendix B for more details). This operator $O$ is then used to obtain both the dissipative and coherent terms as in Eqs. (5) and (6). For each individual instance, we observe the characteristic transition between the fully mixed and pure states and the asymmetric entanglement peak. These features sharpen as the Hilbert space dimension is increased. Therefore, this study demonstrates that sharp $\mathcal{PT}$-symmetry breaking transitions are not restricted to simple systems with a direct classical counterpart and are expected to occur in a large range of systems that obey Eq. (4).

## 6  Generalizations

The symmetry defined in Eq. (4) and the proof about the fully mixed symmetric phase presented in Sec. 4 can be generalized in a straightforward manner to systems with multiple jump operators. For example, we see the same symmetry-breaking effect in a spin system, with Hamiltonian as above, but considering two competing jump operators for each site,

$$c_A^{1,2} = \sqrt{\frac{1 \pm p}{2}} S_A^\pm, \qquad c_B^{1,2} = \sqrt{\frac{1 \mp p}{2}} S_B^\pm. \tag{16}$$

This model, $\mathcal{L}[H; \{\sqrt{\Gamma} c_A^{1,2}\}, \{\sqrt{\Gamma} c_B^{1,2}\}]$, represents two coupled spins, where one is coupled to a positive temperature reservoir while the other is coupled to a negative temperature reservoir. Crucially, this model still obeys the symmetry relation defined in Eq. (4). In Fig. 2(e)–(f) we plot the purity and entanglement negativity for this model with $p = 0.8$. Although in this case the symmetry-broken phase in the limit $\Gamma \to \infty$ is mixed and the transition is shifted to $\Gamma/g = 1/p$, all the signatures of $\mathcal{PT}$-symmetry breaking described above are still clearly visible.

Even more relevant is the fact that all the arguments presented above still apply to systems where parity is complemented by another unitary symmetry, $\mathcal{P} \to \mathcal{P}U$. For example, by choosing $U = e^{i\pi(S_A^x + S_B^x)}$ and a Hamiltonian $H = g(S_A^+ S_B^+ + S_A^- S_B^-)$, we obtain a $\mathcal{PT}$-symmetric quantum system $\mathcal{L}[H; \sqrt{\Gamma} S_A^-, \sqrt{\Gamma} S_B^-]$. While this model contains only loss processes and the occupation numbers $\langle S_A^+ S_A^- \rangle = \langle S_B^+ S_B^- \rangle$ remain symmetric for all ratios of $\Gamma/g$, the Liouvillian respects the symmetry of Eq. (4) with a generalized anti-unitary map

$$\mathbb{PT}(O) = \mathcal{P}UO^\dagger(U\mathcal{P})^{-1}. \tag{17}$$

As a consequence one observes the same transition from a fully mixed to a low-entropy state, as in the spin model discussed above. The symmetry relation in Eq. (4) is thus a powerful tool to identify $\mathcal{PT}$-symmetry breaking effects, even in systems where our naive intuition fails.

# 7 Conclusion

In summary, we have introduced the symmetry relation, Eq. (4), for Liouville operators, which extends the notion of $\mathcal{PT}$ symmetry to bipartite open quantum systems. This definition is consistent with previous examples of linear $\mathcal{PT}$-symmetric quantum systems for which the conventional definition of $\mathcal{PT}$ symmetry is recovered in the limit of large oscillation amplitudes. At the same time the map, $\mathbb{PT}$, in Eq. (3) is completely general and can also be used to study $\mathcal{PT}$ symmetry in highly nonlinear systems or for dissipation processes that have no direct classical counterpart.

In this paper we have mainly focused on the steady state $\rho_0$, which is determined for all parameters by the zero eigenvector of $\mathcal{L}$. In classical systems, $\mathcal{PT}$-symmetry breaking is usually discussed in terms of a transition from purely oscillatory to exponentially damped or amplified dynamics, which is associated with the appearance of exceptional points in the eigenspectrum of the dynamical matrix. This has motivated similar studies of the spectra of Liouville operators, where the appearance of exceptional points [50, 51] or additional symmetries in the complex eigenvalue structure [30, 44, 45] have been discussed. In Fig. 3 we show that the example of two spin $S = 4$ systems, as in Fig. 1(a),(c) above. We show the full Liouville spectrum below and above the transition point in Fig. 3(a)–(b) and the associated dynamics in panels (c)–(d). For the two cases we don't observe any significant differences in the overall eigenvalue structure. Still the evolution of the observables $\langle S_{A,B}^z \rangle$ undergoes the classically expected change from an oscillatory to an overdamped behavior.

This final example confirms our previous conclusion, namely that $\mathcal{PT}$-symmetry breaking is an emergent phenomenon in the dynamics and stationary expectation values of macroscopic observables, which, in general, depend little on individual eigenvalues of $H_{\text{eff}}$ or $\mathcal{L}$. Based on the symmetry in Eq. (4), this effect can now be studied more systematically and used to make physically consistent predictions for real experiments. This will be important, for example, for

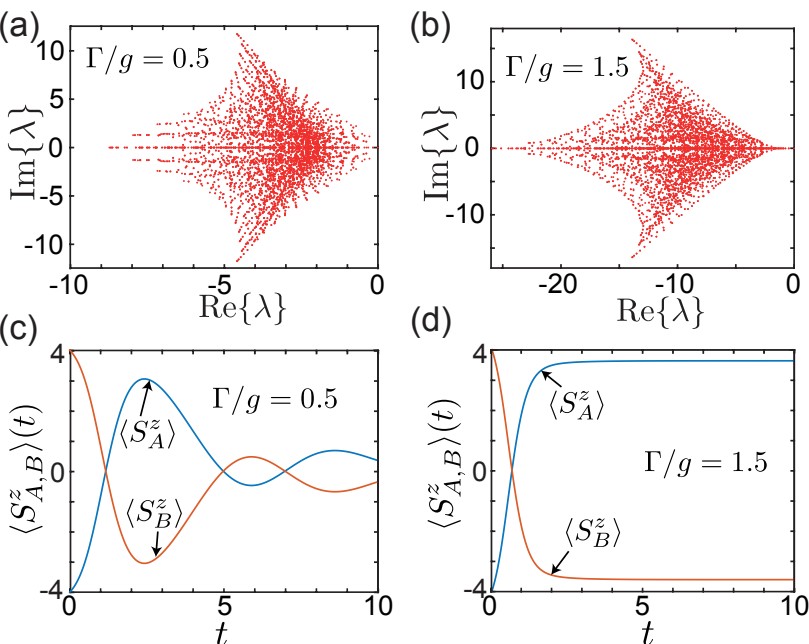

Figure 3: Plot of all complex eigenvalues $\lambda_i$ of the Liouvillian $\mathcal{L}$ for the $\mathcal{PT}$-symmetric spin system introduced in Fig. 1(a) with $S = 4$. In (a) the spectrum is shown below ($\Gamma/g = 0.5$) and in (b) above ($\Gamma/g = 1.5$) the transition point. For the same parameters, (c) and (d) show the corresponding time evolution of the observables $\langle S_{A,B}^z \rangle(t)$, starting from the initial state $\rho(t = 0) = |-S\rangle\langle -S| \otimes |S\rangle\langle S|$.

trapped atoms [52], optomechanics [53] or circuit QED systems [54], where gain and loss but also much more complex dissipation processes can be engineered [55, 56]. Our discussion also shows that there are still many interesting conceptual questions to address. This concerns, in particular, the existence and the nature of the $\mathcal{PT}$-symmetry breaking transition in higher dimensional lattices and interacting many-body system, for which no exact numerical solutions exist.

# Acknowledgements

**Funding information**   This work was supported through an ESQ fellowship (P.K.) and a DOC Fellowship (J.H.) from the Austrian Academy of Sciences (ÖAW) and by the Austrian Science Fund (FWF) through the DK CoQuS (Grant No. W 1210), Grant No. P32299 (PHONED) and Grant No. P32300 (WAVELAND).

# A   Fully symmetric steady state

In this section we detail and extend the proof for the linear stability of the fully mixed symmetric phase in the limit $\Gamma \to 0$ discussed above. As a starting point we write the density operator as

$$\rho = \sum_{n,m} \rho_{n,m}|E_n\rangle\langle E_m|, \tag{18}$$

where $|E_n\rangle$ are the energy eigenstates of $H$, i.e. $H|E_n\rangle = E_n|E_n\rangle$. From the $\mathcal{PT}$-criterion in Eq. (4), we know that $[H, \mathcal{P}] = 0$, and hence we may simultaneously diagonalise the parity operator $\mathcal{P}|E_n\rangle = \zeta_n|E_n\rangle$, where $|\zeta_n|^2 = 1$ without loss of generality.

For $\Gamma = 0$ the fully mixed state, $\rho = \mathbb{1}/d^2$, is a stationary solution of the ME $\dot\rho = \mathcal{L}_H\rho = -i[H, \rho]$, but this is also true for any other diagonal state. Therefore, we make the ansatz $\rho_{n,m} = \delta_{n,m}/d^2 + \delta\rho_{n,m}$ and evaluate the evolution of $\delta\rho_{n,m}$ up to first order in $\Gamma$ [noting that $c_{A,B} \sim O(\sqrt{\Gamma})$],

$$\delta\dot\rho_{n,m} = -\frac{i}{\hbar}(E_n - E_m)\rho_{n,m} + \frac{1}{d^2}\langle E_n|[c_A, c_A^\dagger] + [c_B, c_B^\dagger]|E_m\rangle. \tag{19}$$

We first assume that $E_n \neq E_m$. In this case the elements $\rho_{n,m}$ represent coherences between non-degenerate eigenstates and we obtain

$$\delta\rho_{n,m}(t) \simeq -i\frac{1}{d^2(E_n - E_m)}\langle E_n|[c_A, c_A^\dagger] + [c_B, c_B^\dagger]|E_m\rangle \times \left(1 - e^{-i(E_n-E_m)t/\hbar}\right). \tag{20}$$

Therefore, to lowest order in $\Gamma$ all these off-diagonal elements of the density matrix remain bounded and $|\delta\rho_{n,m}| \to 0$ for $\Gamma \to 0$.

For all other matrix elements with $E_n = E_m$ the coherent evolution vanishes and

$$\delta\dot\rho_{n,m} = \frac{1}{d^2}\langle E_n|[c_A, c_A^\dagger] + [c_B, c_B^\dagger]|E_m\rangle. \tag{21}$$

This results in a linear growth in time, unless the matrix element on the right-hand side is zero. We now make use of the relation

$$\mathcal{P}c_B\mathcal{P}^{-1} = c_A^\dagger, \tag{22}$$

which follows from the $\mathcal{PT}$-symmetry relation for the Liouville operator. Based on this transformation we obtain

$$
\begin{aligned}
\langle E_n|[c_B, c_B^\dagger]|E_m\rangle &= \langle E_n|\mathcal{P}^{-1}\mathcal{P}[c_B, c_B^\dagger]\mathcal{P}^{-1}\mathcal{P}|E_m\rangle \\
&= \langle E_n|\mathcal{P}^{-1}[c_A^\dagger, c_A]\mathcal{P}|E_m\rangle \\
&= -\zeta_n^*\zeta_m\langle E_n|[c_A, c_A^\dagger]|E_m\rangle,
\end{aligned}
\tag{23}
$$

and the evolution equation from above can be written as

$$
\delta\dot{\rho}_{n,m} = \frac{1}{d^2}\langle E_n|[c_A, c_A^\dagger]|E_m\rangle\,(1 - \zeta_n^*\zeta_m).
\tag{24}
$$

In the case of a Hamiltonian $H$ with a non-degenerate spectrum, Eq. (24) only applies to the populations $p_n = \rho_{n,n}$, in which case $|\zeta_n|^2 = 1$ and the right hand side vanishes. This is the result given in the main text.

A bit more care must be taken for Hamiltonians with degeneracies imposed by extra symmetries beyond that generated by $\mathcal{P}$. Even though the populations in a given basis still remain fixed, the build-up of coherences between degenerate levels leads to a deviation from the fully mixed state. If the Hamiltonian has a symmetry, $\mathcal{S}$, such that $[H, \mathcal{S}] = 0$, then the states generated by applying $\mathcal{S}$ to $|E_n\rangle$ are degenerate. From Eq. (24) we see that this leads to a non-identity steady state when two states $|E_n\rangle$ and $|E_m\rangle$ with the same energy have a different parity, $\zeta_n \neq \zeta_m$. However, if $[\mathcal{P}, \mathcal{S}] = 0$ then it is straightforward to see that $\zeta_n = \zeta_m$. Therefore, for the existence of a fully mixed symmetric phase it is in general not enough that $[H, \mathcal{P}] = 0$. In addition, we require that all other non-trivial symmetries of the Hamiltonian also commute with the parity operator, at least within each degenerate subspace.

A simple example where such non-trivial symmetries play a role is the spin model described by the Hamiltonian

$$
H = g(S_A^+ S_B^+ + S_A^- S_B^-)
\tag{25}
$$

and the $\mathcal{PT}$-symmetric ME

$$
\dot{\rho} = -i[H, \rho] + \Gamma\mathcal{D}[S_A^-]\rho + \Gamma\mathcal{D}[S_B^+]\rho.
\tag{26}
$$

This model has a symmetry generated by $\mathcal{S} = S_A^z - S_B^z$ which does not commute with $\mathcal{P}$ and indeed one can show that the steady state for this model has spin-$A$ pointing down and spin-$B$ pointing up independent of the value of $\Gamma/g$.

## B   Random jump operators

In Fig. 2(c)–(d) we calculate the steady state of random $\mathcal{PT}$-symmetric finite dimensional quantum systems. Here we describe how these random models are constructed.

For simplicity we keep the relationship between jump operators and the Hamiltonian as described in the main text. We also wish to ensure that the jump operators have a single dark state, such that in the limit $\Gamma \to \infty$ the purity $P \to 1$.

The procedure we use is then as follows: We first create a random matrix $R$ from the Gaussian orthogonal ensemble (GOE), i.e., a symmetric matrix with real entries which follow a Gaussian distribution [57]. This matrix is then shifted by its lowest eigenvalue such that

$R' = R - \lambda_0 \mathbb{I}$ is positive semidefinite with a guaranteed zero eigenvalue. To obtain the jump operator $O$ we then perform a Cholesky decomposition on the resulting matrix,

$$R' = OO^\dagger, \tag{27}$$

such that $O$ is a lower triangular matrix. Since the Cholesky decomposition for positive semidefinite matrices is not unique, we implement this step by first diagonalizing the random matrix $R'$,

$$R' = UDU^\dagger, \tag{28}$$

with $U$ a unitary matrix and $D = \mathrm{diag}(0, \lambda_1, \ldots, \lambda_{d-1})$, a diagonal matrix where $\lambda_n$ are non-zero eigenvalues. The diagonal matrix $D$ can be decomposed as $D = LL^\dagger$, where only the first super diagonal of $L^\dagger$ is non-zero with $(\sqrt{\lambda_1}, \sqrt{\lambda_2}, \ldots, \sqrt{\lambda_{d-1}})$. As a result the jump operator is

$$O = ULU^\dagger. \tag{29}$$

This procedure of constructing a random jump operator ensures that most of the resulting decay rates are $O(1)$, due to the fact that the spacing between the eigenvalues of $R$ will follow a Wigner surmise distribution $P(\Delta E) \sim \Delta E \exp(-A\Delta E^2)$ [57], meaning that there are very few almost degenerate states. By enforcing $L^\dagger$ to only have non-vanishing elements in the first upper diagonal ensures that it is possible to observe the $\mathcal{PT}$-symmetry breaking transition. This is not guaranteed in general. For example, by decomposing the diagonal matrix $D$ in Eq. (28) in terms of two diagonal matrices $D = \sqrt{D}\sqrt{D}$, the resulting jump operator would be Hermitian and there would be no phase transition since the trivial identity state is always a steady state of such a model.

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
