# Peer review of "Emergence of PT-symmetry breaking in open quantum systems"

_SciPost Physics_

## Round 2 · Referee Report · Anonymous (Referee 1) · 2020-6-14

Strengths

1- Addresses a topical question on quantum dynamics with gain and loss
2- Provides a clear definition of a physically interesting symmetry
3- Illustrates the consequences in meaningful examples

Weaknesses

1- The symmetry employed in this paper is generally known as a particle-hole/charge-conjugation symmetry, which is a well-established symmetry in physics. It should not be confused with a PT symmetry.
2- Some details of the discussion and context, as outlined in the report.

Report

PT symmetric quantum systems were originally introduced as a non-Hermitian generalization of conventional quantum mechanics built upon Hermitian operators. While no fundamentally PT symmetric quantum systems are known, it was later realized that the symmetry occurs naturally in open systems, such as optical systems with gain and loss. However, the underlying dynamics of these systems is classical, so that at this stage the concept was transferred only on a purely phenomenological level.

In this paper the authors consider the question of how to define the PT symmetry in the corresponding quantum setting of these open systems. This question can be pursued along two routes: quantization around a still PT-symmetric mean-field description, or, as here, based on the non-Hermitian dynamical matrix that occurs in a Lindblad master equation, which itself does not obey PT symmetry. The authors address the interesting question if a useful definition of PT symmetry can still be given in the latter case, and suggest that this can be achieved by a transformation that interchanges creation and annihilation operators. By well-selected examples, they demonstrate that this is a physically interesting symmetry, which results in a clear phenomenology.

On the technical level, this is a well-carried out study. However, the paper suffers from a fundamental flaw, namely, that the suggested symmetry has already a well-established place in physics as a particle-hole or charge-conjugation symmetry, which should be seen as clearly distinct from PT symmetry. Indeed, besides being well established for conventional quantum systems, where this particle-hole/charge-conjugation symmetry is crucial for their 10-fold classification, it also features prominently as an independent symmetry (again distinct from any generalized time reversal symmetry such as PT) in recent classifications on non-Hermitian systems. The dressing of this symmetry with the P operator is an interesting twist, but does not change the fundamental nature of the symmetry, in the same way that fundamentally, a PT symmetry is still a time-reversal symmetry.

I consider this a major flaw that has to be addressed before this work can be considered for publication.

Minor remarks: The authors describe PT-symmetry breaking as a transition where eigenvalues become purely imaginary (they mention this twice in the introduction). However, PT-symmetry breaking generally results in complex-conjugated eigenvalues. As a matter of fact, their notion seems to be a symptom of the confusion with the particle-hole or charge-conjugation symmetry, which on the mean-field level gives rise to an anti-PT symmetric Hamiltonian. The latter makes the spectrum symmetric to the imaginary axis; if PT and anti-PT symmetry occur at the same time, eigenvalues are on the real axis, imaginary axis, or form quadruplets in the complex plane.

In the introduction, the point about finite-dimensional Hilbert spaces is unclear; some of the cited PT-quantum optics papers address resonators that have an infinite Hilbert space.

The discussion of the examples could benefit from a brief comparison with the phenomenology of superradiance.

Requested changes

I sketch one possible resolution of this problem; however, there may be other ways to place this work into the proper context:

1- Change the focus of this paper to the consideration of symmetry-breaking transitions in systems with a particle-hole/charge conjugation symmetry. This would indeed be very interesting as quantum noise has been much less studied in this setting.

2-This context would require to expand the references with works on non-Hermitian particle-hole and charge-conjugation symmetry, their interplay with PT symmetry, and their role in present classifications of non-Hermitian systems.

3-If still relevant after such revisions, I would also encourage to mention the mean-field version of the effective Hamiltonian in the Lindblad equation, which can be PT symmetric so that from this perspective there is no conceptual problem to transfer this symmetry to the quantum setting.

4- In the discussion of the examples, the authors should briefly describe the relation or distinction between the phenomenological effects of breaking the symmetry in question, and the general phenomenon of superradiance.

5-Clarify the statement in the introduction about the dimensionality of Hilbert space.

  • validity: low
  • significance: good
  • originality: high
  • clarity: high
  • formatting: excellent
  • grammar: excellent

Author:  Julian Huber  on 2020-08-17  [id 927]

(in reply to Report 1 on 2020-06-14)
Category:
answer to question

First of all, we would like to thank the reviewer for spending the time to carefully read our manuscript and for providing us with detailed and constructive feedback. The reviewer raises one major concern, namely that the PT symmetry we introduce in this paper is just the well-established particle-hole symmetry discussed in symmetry classifications of fermionic systems. The referee’s observation is, indeed, a very interesting one and worth discussing, but it can be shown in terms of a few basic examples that the symmetry introduced in this work is clearly different from particle-hole symmetry:

(i) Ignoring the parity transformation, the operation introduced in Eq. (3) is the Hermitian adjoint. It is true that at the level of a single fermionic operator, taking the Hermitian adjoint corresponds to replacing a particle with a hole. However, in general the two transformations are not the same. For example, a jump operator c=a^\dag a is invariant under the Hermitian adjoint, but it changes into c -> a a^dag under a particle-hole exchange.

(ii) Every (hermitian) Hamiltonian is invariant under H->H^\dag, but not every Hamiltonian is particle-hole symmetric. The physics of PT symmetry breaking described in this work also occurs for systems without particle-hole symmetry, for example, systems with only positive eigenenergies. Here one should also emphasize that in our definition of PT symmetry is only defined for the Liouville operator where the operation in Eq. (3) is applied to the Hamiltonian H and the jump operators separately. This procedure must be distinguished from taking the adjoint of the non-hermitian Hamiltonian H_eff, which is physically not meaningfull.

(iii) In contrast to particle-hole symmetry, the Hermitian adjoint operation cannot be derived from transformations on states, which are the transformations considered in the conventional symmetry classification schemes mentioned by the reviewer.

(iv) Most importantly, our symmetry is completely general and also applies to non-fermionic or non-bosonic systems, where particle-hole symmetry is not even defined. Therefore, the two symmetries cannot be the same.

Let us further remark that currently, there is considerable interest in the symmetries of non-Hermitian Hamiltonians, in particular also on PT symmetric systems. However, such PT symmetric Hamiltonians have no counterpart in the theory of open quantum systems and despite an immense amount of work on “PT symmetric quantum systems”, there is still no physically meaningful definition of exactly what this term means. A key finding of this work is that on a fundamental level, PT symmetry must be considered not as an exact, but as an emergent symmetry. And in order for such an emergent symmetry to exist, the usual T-operation on the Hamiltonian level must be replaced by a Hermitian adjoint operation at the level of a master equation. Even in simple non-interaction fermionic systems, where this adjoint operation essentially reduces to particle-hole exchange, this whole approach and line of argument is very different from other symmetry classifications of non-Hermitian operators or Liouvillians.

In summary, we hope that these arguments will convince the reviewer that this work goes beyond what has been discussed in the literature before and, in particular, that the symmetry introduced in Eqs. (3) and (4) is in general not the same as particle-hole symmetry. Most importantly, this work is not only about identifying a symmetry, but also about its physical consequences.

Reviewer: “The authors describe PT-symmetry breaking as a transition where eigenvalues become purely imaginary (they mention this twice in the introduction). However, PT-symmetry breaking generally results in complex-conjugated eigenvalues. As a matter of fact, their notion seems to be a symptom of the confusion with the particle-hole or charge-conjugation symmetry, which on the mean-field level gives rise to an anti-PT symmetric Hamiltonian. The latter makes the spectrum symmetric to the imaginary axis; if PT and anti-PT symmetry occur at the same time, eigenvalues are on the real axis, imaginary axis, or form quadruplets in the complex plane.”

Our reply: We thank the referee for pointing out our imprecise statement, which we corrected. Our statement was, however, not due to a confusion with other symmetries, but simply related to the fact that in almost all examples of PT symmetric systems in the literature, the eigenvalues become indeed purely imaginary.

Reviewer: “1- Change the focus of this paper to the consideration of symmetry-breaking transitions in systems with a particle-hole/charge conjugation symmetry. This would indeed be very interesting as quantum noise has been much less studied in this setting.”

Our reply: As we explained in detail above, the purpose of our work is to introduce a general definition for PT-symmetric quantum systems and this symmetry is not the same as particle-hole symmetry. We would therefore like to refrain from changing the focus of our study to particle-hole symmetric, which would be a completely different topic.

In the revised manuscript we have added a brief discussion below Eq. (5) to emphasize the difference between our symmetry and particle-hole exchange symmetry.

Reviewer: “This context would require to expand the references with works on non-Hermitian particle-hole and charge-conjugation symmetry, their interplay with PT symmetry, and their role in present classifications of non-Hermitian systems.”

Our reply: In the revised version of the manuscript we added several recent references on the topic of symmetries of non-Hermitian Hamiltonians and master equations for fermionic systems. We now also state explicitly, why our work is different and goes beyond such classification schemes.

Reviewer: “If still relevant after such revisions, I would also encourage to mention the mean-field version of the effective Hamiltonian in the Lindblad equation, which can be PT symmetric so that from this perspective there is no conceptual problem to transfer this symmetry to the quantum setting”

Our reply: We thank the referee for this comment, which addresses a common misconception in this field. The mean field equations of motion can only be PT symmetric for linear harmonic oscillators. For fermions, spins and other quantum systems, this is not the case. In the revised version of the manuscript we now explicitly discuss this point and added a simple example for illustration in the appendix.

Reviewer: “4- In the discussion of the examples, the authors should briefly describe the relation or distinction between the phenomenological effects of breaking the symmetry in question, and the general phenomenon of superradiance”.

Our reply: Superradiance usually refers to the accelerated decay of an ensemble of atoms coupled to a common field mode – an effect that typically does not involve any symmetry breaking. While also in one of our examples we discuss a spin model with collective dissipation, we focus on steady states and the competition between coherent and incoherent processes. We might not fully understand this comment by the reviewer, but we do not see any direct connection between the present work and superradiance.

Reviewer: “In the introduction, the point about finite-dimensional Hilbert spaces is unclear; some of the cited PT-quantum optics papers address resonators that have an infinite Hilbert space. 5-Clarify the statement in the introduction about the dimensionality of Hilbert space.”

Our reply: All previous works on PT symmetric quantum systems that we are aware of have considered bosonic systems with an infinite Hilbert space. For such systems the mean-field equations of motion with PT symmetry can be derived. However, as mentioned above, for generic quantum systems this is not the case. One of the main results of our paper is that our definition of PT symmetry can also be straightforwardly applied to non-bosonic and finite dimensional systems.

We have clarified that we refer to “non-bosonic systems with a finite dimensional Hilbert space, where mean-field approximations are neither justified nor uniquely defined”.

---

## Editorial Decision

resubmitted